# A comprehensive estimation of country-level basic reproduction numbers $R_0$ for COVID-19: Regime regression can automatically estimate the end of the exponential phase in epidemic data

John L. Spouge [ID] *

National Center for Biotechnology Information, National Library of Medicine, National Institutes of Health, Bethesda, Maryland, United States of America

* spouge@nih.gov

## Abstract

In a compartmental epidemic model, the initial exponential phase reflects a fixed interaction between an infectious agent and a susceptible population in steady state, so it determines the basic reproduction number $R_0$ on its own. After the exponential phase, dynamic complexities like societal responses muddy the practical interpretation of many estimated parameters. The computer program ARRP, already available from sequence alignment applications, automatically estimated the end of the exponential phase in COVID-19 and extracted the exponential growth rate $r$ for 160 countries. By positing a gamma-distributed generation time, the exponential growth method then yielded $R_0$ estimates for COVID-19 in 160 countries. The use of ARRP ensured that the $R_0$ estimates were largely freed from any dependency outside the exponential phase. The Prem matrices quantify rates of effective contact for infectious disease. Without using any age-stratified COVID-19 data, but under strong assumptions about the homogeneity of susceptibility, infectiousness, etc., across different age-groups, the Prem contact matrices also yielded theoretical $R_0$ estimates for COVID-19 in 152 countries, generally in quantitative conflict with the $R_0$ estimates derived from the exponential growth method. An exploratory analysis manipulating only the Prem contact matrices reduced the conflict, suggesting that age-groups under 20 years did not promote the initial exponential growth of COVID-19 as much as other age-groups. The analysis therefore supports tentatively and tardily, but independently of age-stratified COVID-19 data, the low priority given to vaccinating younger age groups. It also supports the judicious reopening of schools. The exploratory analysis also supports the possibility of suspecting differences in epidemic spread among different age-groups, even before substantial amounts of age-stratified data become available.

**Data Availability Statement:** All relevant data were obtained from publicly available URLs given in the Supporting Information text file.

**Funding:** This research was supported by the Intramural Research Program of the NIH, National Library of Medicine.

**Competing interests:** The authors have declared that no competing interests exist.

# Introduction

Historically, compartmental models of epidemics derive from the Susceptible-Infectious-Recovered (SIR) model of Kermack and McKendrick [1–3]. In classical compartmental models, epidemics have a distinctive beginning, middle, and end. In the beginning, an epidemic has an exponential phase, growing as $\exp(rt)$, where $r$ is the exponential growth rate. The initial exponential growth rate $r$ is an observable that constrains the basic reproduction number $R_0$ (e.g., [4–6]), the expected number of secondary infections produced by a typical infected individual during its entire period of infectiousness in a completely susceptible population [7]. In the middle of the epidemic, the basic reproduction number $R_0$ provides a baseline for quantifying how dynamic variables like societal responses and the depletion of susceptibles affect epidemic spread. Finally, in the end of the epidemic, $R_0$ constrains the total count of individuals infected by the epidemic as the population returns to a steady state [8–10].

The beginning of an epidemic therefore displays a simplicity lacking during the middle of the epidemic and its dynamic complexities [11]. In fact, the initial exponential phase develops from a fixed interaction between an infectious agent and a population in steady state. In contrast, the dynamic complexities of the middle, particularly societal responses, muddy the interpretation of estimated parameters. Although biology analyzes ever more comprehensive amounts of data, confident human interpretation remains practically useful. With a view to narrowing the complexities muddying interpretation, the demarcation of the exponential phase of an epidemic is therefore a worthwhile aim.

Though arbitrary, a transitional boundary demarcating the exponential phase therefore serves a purpose, and individuals can often concur on such transitional boundaries to within useful accuracies (e.g., see Fig 1 of the Results). The ARRP computer program in sequence alignment demarcates similar transitional boundaries automatically [12]. When estimating statistical parameters for the popular BLAST suite of sequence comparison programs [13–15], Monte Carlo simulations generate data points $(t, y)$ that approach a horizontal line as $t$ tends to infinity (to motivate the discourse, see Fig 2 of [12] for an example). To extract the constant at infinity, ARRP performs a so-called asymptotic regression, a procedure related closely to change-point regression [16]. Change-point regression specifies two statistical models, one for each side of a change-point, and then estimates the unknown position of the change-point. In contrast, asymptotic analysis specifies only a single model, e.g., in the sequence alignment application above, the model for the asymptotic regime near infinite $t$. Conceptually, ARRP moves leftward from infinite $t$, accumulating the data points $(t, y)$ in a list. During the accumulation, ARRP estimates model parameters from each list. Eventually, the residuals for the leftmost points $(t, y)$ display a single sign, as their bias comes to dominate the statistical noise, signaling systematic departure from the model for the asymptotic regime. Formally, ARRP calculates a transitional boundary by minimizing a penalized leftward cumulative sum of normalized residuals over all lists (for further details, see the original article [12]).

Asymptotic regression mimics curve-fitting by the human eye. In many semi-logarithmic plots of COVID-19 cases (e.g., see Fig 1 of the Results or Fig 1 in [17]), the eye can follow the case counts as a line moving rightward out from the logarithmic Y-axis. Eventually, the case counts start to lie systematically below the line, and the rightward "cumulative sum of normalized residuals" strains human credulity in the implicit linear regression. The exponential phase of COVID-19 case curves has a known simple model, before it transitions to a dynamic regime in the epidemic. In the sequence alignment application, the known model applies to the asymptotic (i.e., infinite) regime to the right of a transitional boundary. By contrast, in compartmental models of epidemics, the known model applies near the Y-axis. When applied to epidemics, therefore, asymptotic regression might be more appropriately termed "regime regression", the term used throughout the present article.

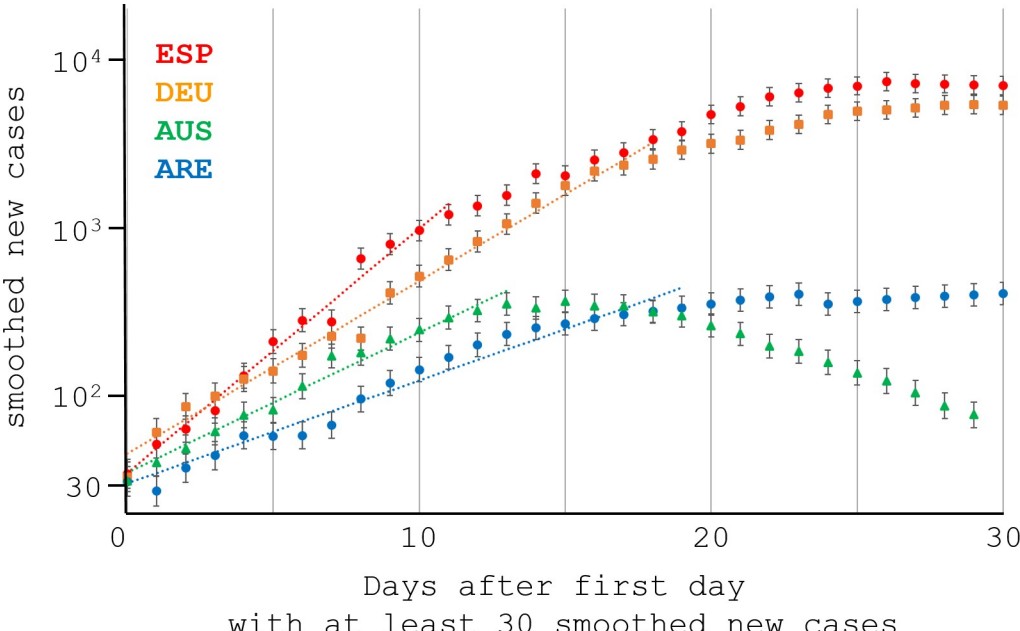

**Fig 1. ARRP regime regression of the initial COVID-19 growth, illustrated with 4 countries.** Fig 1 plots the smoothed new cases on a logarithmic axis against the days since the first day with at least 30 smoothed new cases. Fig 1 shows the exponential phase in 4 countries chosen for illustrative purposes, from top to bottom: Spain EPS (red circles), Germany DEU (orange squares), Australia AUS (green triangles), and United Arab Emirates ARE (blue circles). In the notation of the subsection "Regime regression" in the Materials and Methods, the points plotted are $(t, y\pm\varepsilon)$. Each dotted line displays an ARRP regime regression, described in the Materials and Methods. The largest X-coordinate of each line indicates the estimated transitional boundary terminating the exponential phase of the epidemic: Day 11 (EPS), 18 (DEU), 13 (AUS), or 19 (ARE). In the same order, the 4 countries yielded slopes $r$ from the semi-log graph in Fig 1: 0.34, 0.24, 0.19, or 0.14. Generally, ARRP and visual estimation locate similar post-exponential transitions, particularly for: (1) a large regression slope (ESP or DEU); (2) large smoothed case numbers $y$, with concomitantly small errors $\varepsilon$ (ESP or DEU); (3) a large fall in cases immediately after the post-exponential transition, possibly due to lockdowns or other non-pharmaceutical interventions (AUS); and (4) relatively little unmodeled, systematic noise in the data (AUS).

Regime regression does not require human supervision. For each country with pertinent COVID-19 data, the present article uses ARRP to demarcate a transitional boundary. ARRP then performs a weighted linear regression to estimate the initial exponential growth rate $r$ as a mean with a standard error mean. Throughout, the present article makes an "iso-distributional assumption", that the random generation time of SARS-CoV-2 infection has (approximately) the same distribution the world over, regardless of who infected whom [18]. The generation-time distribution then determines $R_0$ for every country [5], using $r$ according to the "exponential growth method" [19].

Using stronger assumptions, the Prem contact matrices [20] also determine $R_0$ [21]. The Prem contact matrices stratify the contact rates in 152 countries into 16 different age-groups. Each Prem contact matrix corresponds to a 16-by-16 next-generation matrix whose elements stratify according to age the expected number of secondary infections from a primary infection in a completely susceptible population [22]. Hilton and Keeling [21] noted that for each country, the Prem contact matrix determines $R_0$ if the Prem contact matrix is proportional to the next-generation matrix, with a fixed, known constant of proportionality.

The present article makes the iso-distributional assumption and estimates the initial exponential growth rates $r$ for 160 countries by applying the ARRP program for regime regression to unstratified COVID-19 case data. It then compares $R_0$ estimated from the growth rates $r$ to $R_0$ estimates derived from the Prem contact matrices. The primary aim of the present article is

to present regime regression as a tool in the compartmental modeling of epidemics. To focus its aim, therefore, it avoids analyzing age-stratified COVID-19 data. It does, however, explore tangentially the possibility that certain age-groups, notably children and adolescents, contributed negligibly to the COVID-19 epidemic by examining how *ad hoc* deletion of age strata from the Prem matrices affected the corresponding estimates of $R_0$.

## Materials and methods

The Supplementary Information gives complete URLs for all files downloaded in this study.

### UN ISO 3166–1 alpha-3 country codes

Throughout the study, 3-letter UN ISO 3166-1 alpha-3 Country Codes from UNSD — Methodology.csv (downloaded from the United Nations 2020-12-07) encoded relevant countries. Because irrelevant columns had formatting errors, Python extracted the relevant data frame only from specific columns within the file.

### COVID-19 data

Our World in Data (OWID) at Oxford University provided COVID-19 data for countries across the world in owid-covid-data.json (downloaded 2020-12-07 from OWID). The file identified its countries with the 3-letter ISO 3166 codes above, and for dates in 2020, it included new COVID-19 cases smoothed over 7 days as running averages. For convenience, we call these running averages "smoothed new cases".

### Regime regression

Data selection generally followed the procedure on 2020-12-09 underlying the OWID graph "Daily new confirmed cases of COVID-19". For each country ($k$) in the OWID COVID-19 data, a script discarded data until the first date with 30 smoothed new cases, which it designated Day $t^{(k)}$ = 0. Then, the script then captured the smoothed new cases on Days $t^{(k)}$ = 0,1,2,... until $T^{(k)}$, the first day either lacking data, or having 1.0 or less smoothed new cases. Regression requires at least 2 points, so the script discarded any country ($k$) with $T^{(k)} \leq 1$. The standalone program ARRP Version 1.1 (within ARRP_1.1.zip downloaded from the National Center for Biotechnology Information 2020-07-14) performed regime regression. ARRP had its default settings except for its option -include left, which forced it to include $t^{(k)}$ = 0 in the exponential phase. For each country, ARRP estimated the transitional boundary terminating the exponential phase of the epidemic, when the smoothed case numbers first displayed a cumulative bias exceeding estimated random errors.

ARRP has 3 columns in its input files. For each country ($k$), they were: (1) $t$, where $0 \leq t = t^{(k)} < T^{(k)}$; (2) $y = \ln Y$, the natural logarithm of the smoothed number of new cases $Y = Y^{(k)}(t)$; and (3) $\varepsilon$, the estimated error in $y$.

To estimate $\varepsilon$, consider a Poisson variate $X$ with mean $\mathbb{E}X = \lambda$ and variance $\sigma^2(X) = \lambda$. The Supplementary Information presents approximations $\mathbb{E}\ln X \approx \ln\lambda$ and $\sigma(\ln X) \approx \lambda^{-1/2} \approx X^{-1/2}$ for $\lambda \geq 30$ (see Fige 5 in [23] for direct numerical support of the approximations). Because of incomplete reporting and subclinical COVID-19 cases [24], the unsmoothed count of new cases is likely sparsely sampled from all cases, so it is reasonably approximated by a Poisson variate. If $Y$ under consideration were the unsmoothed new case count, therefore, the estimate $\varepsilon = \sigma(\ln Y) \approx Y^{-1/2}$ would be reasonable. Preliminary visual inspection of graphs using smoothed new cases showed that under the Poisson error $\varepsilon \approx Y^{-1/2}$, however, ARRP consistently over-estimated the duration of the exponential phase. ARRP often included in it an obvious downward curvature away from exponential growth, because if noise estimates are

inflated, ARRP becomes insensitive to systematic biases. The smoothed new cases $Y$ are running averages over 7 days, however, so ARRP actually used the reduced error $\varepsilon \approx (Y/7)^{-1/2}$, as justified in the Supplementary Information. ARRP estimates the initial slope of the $(t, y \pm \varepsilon)$-plot and provides an estimate with error $r \pm \Delta r$ for the exponential growth rate, where the initial COVID-19 epidemic growth is proportional to the exp($rt$).

## The exponential growth method [19]

Under the iso-distributional assumption, the initial exponential growth rate $r$ and the basic reproduction number $R_0$ satisfy the equation

$$R_0 M(-r) = R_0 \mathbb{E}[\exp(-rT)] = 1, \qquad (1)$$

where $\mathbb{E}$ denotes mathematical expectation; $T$, the random generation time for the infectious disease; and $M(s) = \mathbb{E}[\exp(sT)]$, the moment-generating function of $T$ [5]. If the generation time $T$ has a gamma distribution with mean $\mu = \mathbb{E}T$ and dispersion parameter $\kappa = \sigma^2(T)/\mu^2$, then $M(s) = \mathbb{E}[\exp(sT)] = (1 - s\mu\kappa)^{-1/\kappa}$. Eqs (5) and (6) in the Supplementary Information show that a linear approximation, any error $\Delta r$ in $r \pm \Delta r$ propagates to the estimated $R_0$ as

$$R_0 = \frac{1}{M(-r \pm \Delta r)} = (1 + r\mu\kappa)^{1/\kappa}\left(1 \pm \frac{\mu}{1 + r\mu\kappa}\Delta r\right). \qquad (2)$$

To estimate the distribution of the generation time $T$, the serial time has the same mean $\mu$ and is relatively easy to estimate. A fixed effects model in a meta-analysis estimated the mean serial time in COVID-19 infection as $\mu = 5.40$ [25]. The serial time usually has a larger variance than the generation time [26], but very few articles give separate estimates for generation and serial intervals. One such article estimated the standard deviation of the generation time in Singapore as $\sigma = 1.72$, however [27]. The tab "R0 vs gen time (mu, kappa)" in the S1 File calculates $R_0$ from $\mu$ and $\kappa$ numerically, showing that $R_0$ is insensitive to plausible errors in $\sigma$. For example, our chosen estimates $\mu = 5.40$ and $\sigma = 1.72$ lead to $R_0$ estimates with no more than 20% error for $5.19 < \mu = 5.40 < 5.61$ and $1.50 < \sigma = 1.72 < 2.50$.

For fixed $r$, elementary calculus shows that $R_0$ is an increasing function of $\mu$ and a decreasing function of $\sigma$. All estimates of $R_0$ therefore change in in the same direction in response to errors in $\mu$, and likewise to errors in $\sigma$, enhancing the robustness of scientific conclusions against perturbations in $\mu$ and $\sigma$.

## Preliminary regime regressions used unsmoothed new cases

OWID only recently added data with smoothed new cases, presumably because the unsmoothed data had obvious reporting biases, e.g., due to the day of the week. With large unmodeled errors in unsmoothed data, ARRP truncated some of the estimated exponential phases prematurely, and some of its $R_0$ estimates then exceeded the current consensus that $R_0 \leq 6$ (e.g., [21, 28–30]). Even with the unsmoothed data, all anomalous $R_0$ estimates disappeared, when scripts discarded any exponential phase with a duration of less than 7 days. Smoothed new cases eliminated the need for the 7-day threshold, or indeed, any other threshold for minimum duration of the exponential phase.

## Estimation of the basic reproduction number $R_0$ from Prem contact matrices

For 152 countries ($k$), a 16-by-16 Prem contact matrix $\mathbf{C} = \|C_{a,b}\| = \mathbf{C}^{(k)}$ stratifies their population into ½-decades by age up to 80 years [20], with the elements $C_{a,b}$ estimating an effective contact rate by which a person in Stratum $b$ can transmit infectious diseases to a person in

Stratum $a$. The contact rate $C_{a,b}$ influences $R_{a,b} = R_{a,b}^{(k)}$, the average number of secondary infections in Stratum $a$ caused in a completely susceptible population by a single infected individual in Stratum $b$ [22]. Let the spectral radius $\rho(\mathbf{R})$ denote the dominant (largest non-negative) eigenvalue of the next-generation matrix $\mathbf{R} = \|R_{a,b}\|$. The basic reproduction number $R_0 = R_0^{(k)} = \rho(\mathbf{R})$ is the average number of secondary infections caused by a typical infected individual in a completely susceptible population [7].

Define the basic contact rate $C_0 = C_0^{(k)} = \rho(\mathbf{C})$, and make a strong homogeneity assumption, that a population is (probabilistically) homogeneous in every property relevant to an infectious disease, with the single exception of the contact rates in the matrix $\mathbf{C}$. Then within the population, susceptibility, infectiousness, and disease characteristics like generation time, e.g., do not vary systematically, so the matrices $\mathbf{R}$ and $\mathbf{C}$ are proportional [21]. Proportionality implies that $R_0 \propto C_0$, i.e., there exists some constant $\alpha$, such that $R_0^{(k)} = \alpha C_0^{(k)}$ for each country ($k$).

The Prem_2020 Contact Matrices (downloaded 2020-07-14 from PLoS) contained the Prem contact matrices for all locations (home, work, school, and other) in MUestimates_all_locations_1.xlsx and MUestimates_all_locations_2.xlsx. An online site converted the Excel files to JSON, so another script could separate the tabbed Prem contact matrices into multiple files, denoted [3-letter ISO 3166 country code].csv by country. The statsmodels routine (Version 0.12.1) from NumPy in Python calculated dominant eigenvalues like $C_0 = C_0^{(k)}$ for each country ($k$).

Now, weaken the strong homogeneity assumption, so that homogeneity holds as before, with the sole exception that only certain age-strata $A$ transmit the disease. Consider the principal submatrix $\mathbf{C}_A$ of the Prem contact matrix $\mathbf{C}$ formed by elements $C_{a,b}$ ($a,b \in A$), i.e., elements whose row index $a$ and column index $b$ both lie in $A$. Define a modified basic contact rate $C_{0,A} = C_{0,A}^{(k)} = \rho(\mathbf{C}_A)$. Under the modified homogeneity assumption, $R_0 \propto C_{0,A}$, i.e., there exists some constant $\alpha$, such that $R_0^{(k)} = \alpha C_{0,A}^{(k)}$ for each country ($k$). The Supplementary Information discusses the interpretation of the modified basic contact rate $C_{0,A}$ at greater length.

## Results

All graphical results also appear numerically in the S1 File.

The figures below display choropleths of the world, colored according the duration of the exponential phase (Fig 2), the exponential growth rate $r$ (Fig 3), and the basic reproduction number $R_0$ (Fig 4). (The S1 File also contains a choropleth for the exponential doubling time $t_d = \ln 2/r$, but the distribution of $t_d$ diverges more from a uniform distribution than the distribution of $r$, so because choropleths are linearly-colored, the choropleth for $t_d$ is less informative than the choropleth for $r$.)

Gray regions in the choropleths either lacked COVID-19 data (e.g., Turkmenistan TKM; or People's Republic of Korea PRK) or did not meet the initial threshold of 30 smoothed new cases on two consecutive days, as described in the Materials and Methods (e.g., Tanzania TZA; Congo COG; Chad TCD; Togo TGO; Eritrea ERI; Bhutan BTN; or Laos LAO).

The choropleths therefore provide a comprehensive quantitative visual summary of the exponential phase of COVID-19 for countries worldwide. Many qualitative features are of course well known, such as the aggressive initial spread of COVID-19 in industrialized countries and the slow initial spread in Africa. Of particular interest later, the largest initial slopes as estimated here were Spain ESP (0.34 day$^{-1}$), Iran IRN (0.34 day$^{-1}$), United States USA (0.30 day$^{-1}$), Turkey TUR (0.29 day$^{-1}$), and Germany DEU (0.24 day$^{-1}$), all readily visible in Fig 3, with the possible exception of Germany DEU. The basic reproduction number $R_0$ in Fig 4 is an

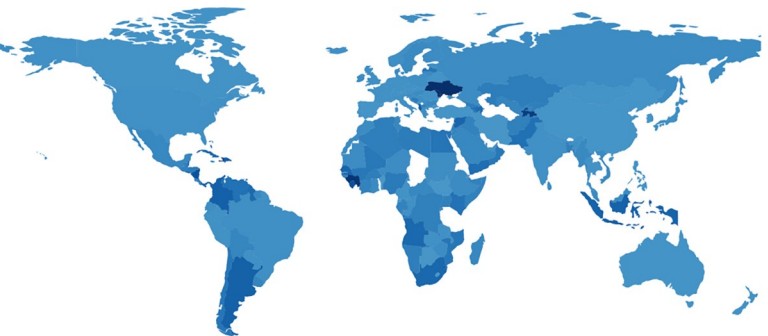

**Fig 2. The duration of the initial exponential phase of COVID-19 growth in different countries.** Fig 2 shows the duration of the initial exponential phase of COVID-19 growth, as computed by the regime regression illustrated in Fig 1. Fig 2 colors countries darkening linearly from lightest blue (4 days for Sao Tome and Principe, STP) up to darkest blue (256 days for Ukraine, UKR). Python GeoPandas generated Fig 2.

increasing function of the exponential growth rate *r* in Fig 3, so it emphasizes the same numerical contrasts as Fig 3, but perhaps more strikingly.

Table 1 provides estimates of the initial exponential growth *r* for four European countries from [19]; Table 2, the corresponding estimates from regime regression. The estimates in Table 1 represent the best model fit of case counts to an exponential curve over all possible start and end dates for the exponential phase. Table 1 displays the countries, the start and end dates of the best fit for each country, and the corresponding estimate of *r*. The countries are in descending order of their slopes as estimated by regime regression in Table 2.

Table 2 below includes Iran (for later purposes) and all countries in Table 1, but the estimates are from regime regression. In Table 2, the estimated errors in the slope (not shown) are all between about ±0.01 and ±0.02. The exponential growth method of estimating $R_0$ requires the characterization of the generation time distribution, which has improved between the publication of [19] and the present article. Unlike Table 1, therefore, Table 2 includes estimates of $R_0$.

The slopes for Spain and Italy in Table 2 accord reasonably well with Table 1, and the corresponding $R_0$ estimates here (5.35 and 2.82) also accord reasonably well with $R_0$ estimated elsewhere (6.00 and 3.60). The slopes for France are in discord, but $R_0$ estimated here (2.69)

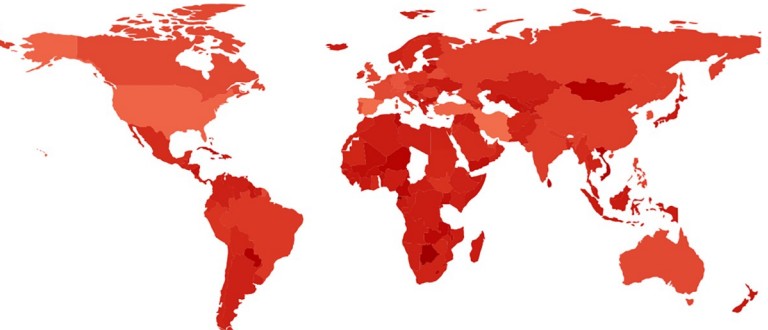

**Fig 3. The exponential growth rate *r* in the exponential phase of COVID-19 in different countries.** On one hand, Fig 2 shows the duration of the initial exponential growth of COVID-19 as computed by regime regression; on the other, Fig 3 shows computed initial exponential growth rate *r*. Fig 3 colors countries by darkening linearly from lightest red (*r* = 0.34 days$^{-1}$ for Spain ESP) down to darkest red (*r* = −0.30 days$^{-1}$ for Equatorial Guinea GNQ). Predictably, the duration characterizing the initial exponential growth in Fig 2 is negatively correlated with the slope in Fig 3 (Pearson correlation coefficient = −0.33). Python GeoPandas generated Fig 3.

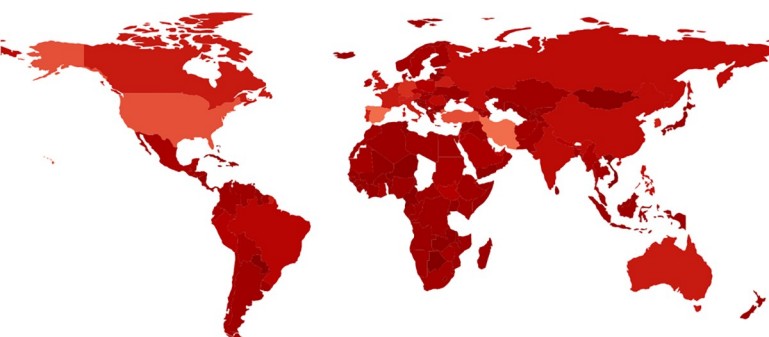

**Fig 4. The estimated basic reproduction number $R_0$ of COVID-19 in different countries.** Eq (2) estimates the basic reproduction number $R_0$ as an increasing function the exponential growth rate $r$. On one hand, Fig 3 shows the initial exponential growth rate $r$; on the other, Fig 4 shows the corresponding basic reproduction number $R_0$. Fig 4 colors countries by darkening linearly from lightest red ($R_0$ = 5.35 for Spain ESP) down to darkest red ($R_0$ = 0.17 for Equatorial Guinea GNQ). Python GeoPandas generated Fig 4.

accords well with $R_0$ estimated for France elsewhere (2.90). Germany presents the main discord in Table 2, both in slope (0.24 vs 0.34 in Table 1) and in the estimated $R_0$ (3.37 vs the estimate of 5.46 cited from Table 3 in [29]). Accordingly, Germany provides an instructive case for closer examination in the Discussion.

Under the strong homogeneity assumptions in the Materials and Methods, Fig 5 would display a linear relationship through its origin, which it palpably does not. The legend in the upper right of Fig 5 identifies the region containing each country with a marker of characteristic color and shape. The dominant eigenvalues $C_0$ of the Prem contact matrices therefore appear to separate countries by regions better than they introduce any ordered pattern into the $R_0$ estimates on the Y-axis.

In contrast to Figs 5 and 6 has a relatively simple compact cluster structure, arguably the simplest structure of all similar figures in the Supplementary Information derived from principal submatrices of the Prem contact matrices. Moreover, an Excel unweighted linear regression through (0, 0) of the 5 topmost points (DEU, TUR, USA, ESP, and IRN) yields $R^2$ = −0.051 (slope 0.30) in Fig 5 but $R^2$ = 0.860 (slope 0.44) in Fig 6. (Regression without an intercept can yield a negative $R^2$.) For descriptive purposes, therefore, the unweighted $R^2$ quantitatively reinforces the impression that the 5 topmost points have moved closer to the ideal of a straight line through (0, 0). Table 2 contains citations that estimated $R_0$ in various countries. The citations accord well with $R_0$ for Spain and Iran in Figs 5, 6. The Discussion examines the discordant $R_0$ for Germany. Articles estimating $R_0$ for the whole of the United States or the whole of Turkey were not found.

## Discussion

Not every epidemic has an initial exponential phase [34–36]. If it does, however, regime regression can estimate the end of the exponential phase automatically and reproducibly, helping to

**Table 1. Initial exponential growth $r$ from estimates in bold from Table 1 of [19].**

| Country | Start | End | Slope |
|---------|-------|-----|-------|
| Spain | 19-Feb | 9-Mar | 0.30 |
| Germany | 21-Feb | 9-Mar | 0.34 |
| Italy | 23-Feb | 9-Mar | 0.21 |
| France | 23-Feb | 9-Mar | 0.34 |

**Table 2. Initial exponential growth _r_ estimated by regime regression.**

| Country | Start | End | Slope | $R_0$ | $R_0$ (Citation) |
|---|---|---|---|---|---|
| Spain | 5-Mar | 16-Mar | 0.34 | 5.35±0.34 | 6.00 [29] |
| Iran | 27-Feb | 7-Mar | 0.34 | 5.35±0.45 | 4.70 [31] |
| Germany | 4-Mar | 22-Mar | 0.24 | 3.37±0.11 | 5.46 [29] |
| Italy | 24-Feb | 15-Mar | 0.20 | 2.82±0.34 | 3.60 [32] |
| France | 4-Mar | 23-Mar | 0.19 | 2.69±0.08 | 2.90 [33] |

estimate initial parameters like $R_0$ and possibly the date when non-pharmaceutical interventions like lockdowns first took effect [37]. Some important epidemic parameters, e.g., transmission rates β or recovery rates γ, may be more conveniently estimated after the end of the exponential phase, in which case the corresponding truncation of epidemic data has limited utility. Implicitly, however, estimates of such dynamic parameters like β or γ may fluctuate because of societal responses, temporary biases in reporting, etc. Regime regression can truncate epidemic data, thereby eliminating dynamic complications that can muddy the interpretation of estimates of initial parameters like $R_0$ [11].

The exponential phase may suffer its own dynamic complications, e.g., reporting and testing capacity may fluctuate during the exponential phase. Estimates of the initial exponential growth are unlikely to suffer greatly, however, unless the dynamism shows wide fluctuations during the initial exponential phase. If fluctuations are narrow, they do not contribute substantially to the exponent, only to its prefactor. One should note, however, that the initial exponential phase may not be representative of the societal steady state, particularly if the society has prepared for the coming epidemic. Countries may vary substantially in their preparations, with corresponding distortions of their societal steady states.

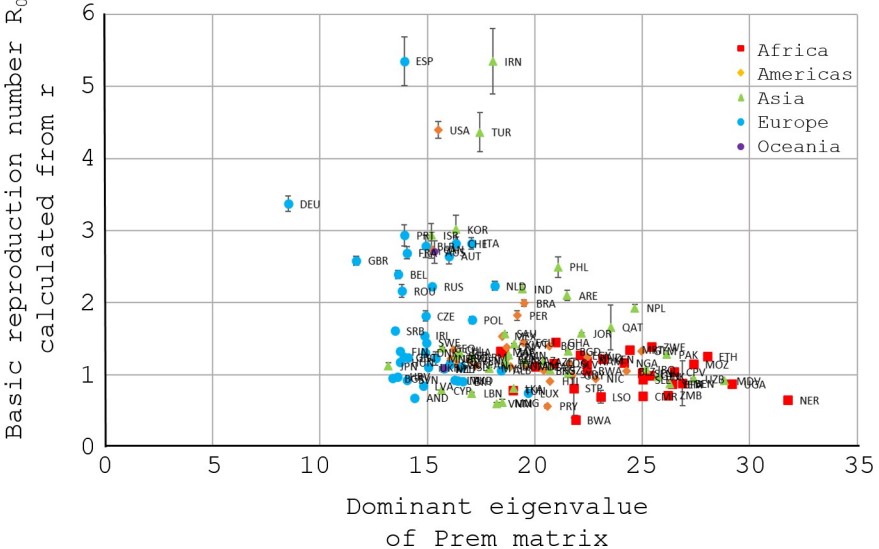

**Fig 5. $R_0$ from Fig 4 vs the dominant eigenvalue $C_0$ of the Prem contact matrix.** Fig 5 plots $R_0$ from Eq (2) with error against the basic contact rate, the dominant (largest non-negative) eigenvalue $C_0$ of the Prem matrix [20]. The 3-letter codes label the countries (the S1 File also gives the same information as Fig 5, but in tabular form). Red squares indicate countries in Africa; yellow diamonds, in the Americas; green triangles, in Asia; blue circles, in Europe; purple squares, in Oceania (i.e., in AUS & NZL). As in the Materials and Methods, consider an idealized world whose population is (probabilistically) homogeneous in every property relevant to COVID-19, with the single exception of having a different Prem contact matrix **C** for each country. In such an idealization, Fig 5 would display a straight line through (0, 0).

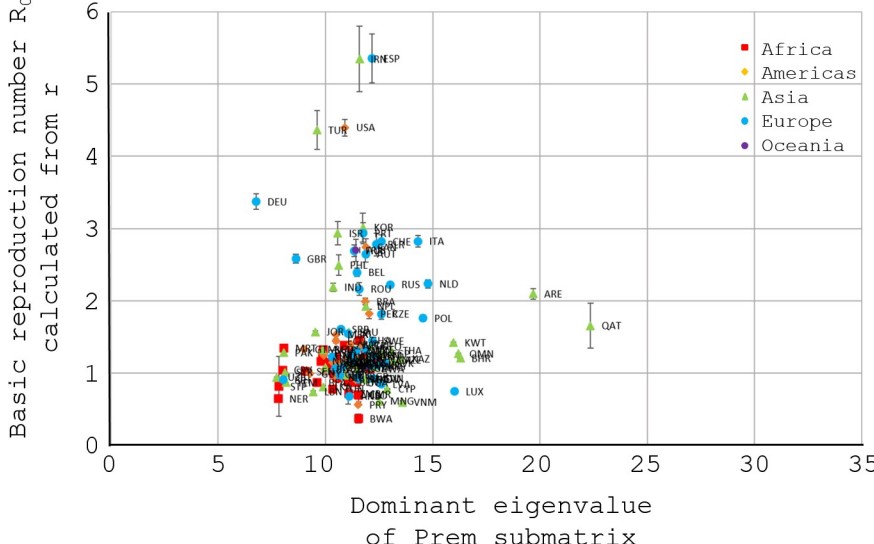

**Fig 6. $R_0$ from Fig 4 vs the dominant eigenvalue of a principal submatrix of the Prem contact matrix.** Fig 6 is like Fig 5 and plots $R_0$ from Eq (2) with error, this time against a modified basic contact rate, the dominant (largest non-negative) eigenvalue of a principal submatrix of the Prem contact matrix [20]. The submatrix deletes rows and columns from the Prem contact matrix if they correspond to age-groups up to 20 years. Consequently, Fig 6 has the same interpretation as Fig 5, except that people up to 20 years old do not contribute to the dominant eigenvalue of the submatrix or, equivalently, to the corresponding epidemic model. The deletion decreases the dominant eigenvalue of Prem contact matrix, moving all points in Fig 5 horizontally and leftward to their positions in Fig 6. Older populations like Germany DEU (leftmost blue circle) undergo less movement from Fig 5 to Fig 6 than younger populations like those in Africa (red squares) The 5 topmost points (DEU, TUR, USA, ESP, and IRN) move noticeably closer to the idealization mentioned in the legend of Fig 5, a straight line through (0, 0).

In COVID-19, primary infections vary widely in the secondary infections they induce [38, 39]. An epidemic like COVID-19 may therefore have a pre-ignition phase where it sputters, with repeated introductions and extinctions in a population, before igniting in the exponential phase that begins an epidemic. Following OWID, therefore, the present article took the first day with 30 cases after 7-day smoothing as the *ad hoc* beginning of a country's COVID-19 epidemic.

OWID also smooths new cases over 7 days, reducing unmodeled noise due to weekly rhythms in reporting. Smoothing improved the robustness of estimates from regime regression. Increased noise, including unmodeled noise, tends to increase the estimated duration of the exponential phase slightly, particularly for slow initial growth (e.g., see ARE in Fig 1). Typically, if the estimated duration is inflated, it biases the estimate of the exponential growth $r$ downward slightly. In its original application (Monte Carlo simulation in sequence alignment), regime regression did not encounter unmodeled noise. In the present application, unmodeled noise probably biased $R_0$ estimates down slightly. Regime regression is automatic, however, so it permits systematic technical improvements. In principle, further work can easily adapt regime regression to unmodeled noise by down-weighting outliers and replacing intermediate linear regressions with robust regressions [40].

Regime regression provides a principled automatic fit for the initial exponential growth, but its estimated $R_0$ for Germany showed a substantial discordance with a competing estimate (3.37 vs 5.46 in Table 2). Note that: (1) $R_0$ is an average, so its estimation should not be unduly influenced by epidemic fluctuations like early superspreading events, and (2) the $R_0$ estimate for Germany (DEU) is based directly on the smoothed case counts displayed in Fig 1. Fig 1 and Table 2 show that Day 0, when smoothed case counts first exceeded 30, was 4-Mar for

Germany. To ease reading, we translate all dates for Germany into "Days" on the X-axis in Fig 1.

From Table 1, the best exponential fit for Germany [19] occurred from Day -12 to Day 5. Possibly, the best exponential fit may have truncated the exponential phase prematurely, before case counts grew enough to display average behavior dependably. In Fig 1, Days 9 to 16 appear linear, with large smoothed case counts from about 500 to 2000. A different study [37] found three epidemic change-points for Germany at Day -1, Day 15, and Day 19 (3-Mar, 19-Mar, and 23-Mar), so the interval from Day -1 to Day 15 may correspond approximately to our estimated exponential phase in Table 2, from Day 0 to Day 18.

Notably, a super-spreading event occurred in a Berlin nightclub on Day -5 (28-Feb) [41], suggesting the possibility that random fluctuations unduly inflated estimates of the mean parameter $R_0$ in early case counts from Germany (examine Fig 1 near Day 0). Fig 5 and the exploratory analysis of Fig 6 also support an expectation that Germany should have a substantially lower $R_0$ than Spain, because the Prem contact matrices yield much lower basic contact rates for Germany than for Spain.

On the other hand, simulations elsewhere suggested that the exponential growth method may seriously and systematically underestimate $R_0$, specifically because it discounts early super spreading [17]. Fig 1 displays around Day 0 for Germany the considerable effect of early super spreading but in the opposite direction, with early fluctuations unduly and temporarily inflating the apparent exponential growth. The effect of early super spreading on $R_0$ estimates therefore requires further investigation.

To simplify the initial presentation of regime regression in epidemiology, the present article has generally avoided modeling population inhomogeneities. As an important example of inhomogeneities in epidemics, however, asymptomatic cases may transmit COVID-19 less than symptomatic or presymptomatic cases [42, 43]. Generally, children display fewer symptoms and probably transmit COVID-19 less readily than adults [44]. By its structural simplicity relative to Fig 5 and similar graphs in the Supplementary Information, Fig 6 suggests that the top five initial exponential growth rates of COVID-19 in various countries (Spain EPS, Iran IRN, United States USA, Turkey TUR, and Germany DEU) are more readily explained if subpopulations under 20 years old contributed much less to the initial epidemic transmission than their elders. Fig 6 therefore supports suggestions elsewhere that age-specific heterogeneities beyond contact structure were important in the initial spread of COVID-19 [21].

Beyond the top five, the remaining countries in Fig 6 fall into two or three clusters lacking a ready explanation. The Prem contact matrices were extrapolated from eight European countries to 152 nations, so the clusters may reflect the extrapolation. They also may reflect, e.g., disparate reporting biases or other dynamic factors outside the societal steady state, including pre-adaptation to the coming epidemic. To reach definite conclusions, many studies narrow their subject by selecting the countries under study, e.g., [24]. By contrast, the design of Figs 2–6 deliberately included as many countries as possible, to provide a global overview.

The exploratory analysis in Fig 6 with principal submatrices of the Prem contact matrices adds to the information in Fig 5. Populations in different countries have different age-structures, so their age-specific contact rates vary. The principal submatrices therefore permitted an age-stratified analysis, even without any age-stratified data specific to COVID-19. In principle, the analysis could have been refined by pre- and post-multiplying each Prem contact matrix by diagonal matrices representing age-stratified susceptibility, symptomatic fractions, and infectivity, in the same spirit as fig 1b in [24]. Without specific age-stratified data, however, a refined analysis seems premature, and the use of age-stratified data goes beyond the purview of the present paper.

In summary, vaccination, viral mutants with increased infectiousness, and population heterogeneity all conspire to reduce the predictive utility of $R_0$ [11, 45], but typically $R_0$ and its judicious interpretation remain helpful throughout an epidemic. Fig 4 shows that the $R_0$ estimates for COVID-19 vary across countries, perhaps to a surprising degree. The countries' populations also vary widely in age-structure, so the $R_0$ estimates permitted an exploratory analysis with principal submatrices of the Prem contact matrices, suggesting that age-groups under 20 years might not have promoted the initial exponential growth of COVID-19 as much as other age-groups. The exploratory analysis therefore supports tentatively and tardily, but largely independently of age-stratified data, the vaccination strategy giving low priority to younger age groups. It also supports the judicious reopening of schools, a topic of current concern [46]. It also supports the possibility of suspecting differences in epidemic spread among different age-groups, even before substantial amounts of age-stratified data become available, much as others have already suggested [20, 21].

## Supporting information

**S1 Readme. This file provides an index for the supporting information.**
(DOCX)

**S1 Methods. This file includes additional methods and publicly available URLs for all data and code.**
(DOCX)

**S1 File. This file includes all figures in tabular form.**
(XLSX)

## Author Contributions

**Conceptualization:** John L. Spouge.

**Formal analysis:** John L. Spouge.

**Investigation:** John L. Spouge.

**Methodology:** John L. Spouge.

**Software:** John L. Spouge.

**Validation:** John L. Spouge.

**Visualization:** John L. Spouge.

**Writing – original draft:** John L. Spouge.

**Writing – review & editing:** John L. Spouge.

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
