## [Decision Letter · Decision Letter 0]

26 Apr 2021

PONE-D-21-06056

A comprehensive estimation of country-level basic reproduction numbers R0 for COVID-19: regime regression can automatically estimate the end of the exponential phase in epidemic data

PLOS ONE

Dear Dr. Spouge,

Thank you for submitting your manuscript to PLOS ONE. After careful consideration, we feel that it has merit but does not fully meet PLOS ONE’s publication criteria as it currently stands. Therefore, we invite you to submit a revised version of the manuscript that addresses the points raised during the review process.

Specifically, while both reviewers agree in the manuscript is technically sound, reviewer #1 points out that the manuscript would be improved by describing potential applications to future pandemics and reviewer #2 thinks greater attention should be given to the potential limitations of the study. 

We look forward to receiving your revised manuscript.

Kind regards,

Nicholas S. Duesbery, PhD

Academic Editor

PLOS ONE

Journal Requirements:

2. We note that Figures 2-4 in your submission contain map images which may be copyrighted. All PLOS content is published under the Creative Commons Attribution License (CC BY 4.0), which means that the manuscript, images, and Supporting Information files will be freely available online, and any third party is permitted to access, download, copy, distribute, and use these materials in any way, even commercially, with proper attribution. For these reasons, we cannot publish previously copyrighted maps or satellite images created using proprietary data, such as Google software (Google Maps, Street View, and Earth). For more information, see our copyright guidelines: http://journals.plos.org/plosone/s/licenses-and-copyright.

2a, You may seek permission from the original copyright holder of Figures 2-4 to publish the content specifically under the CC BY 4.0 license. 

2b.           If you are unable to obtain permission from the original copyright holder to publish these figures under the CC BY 4.0 license or if the copyright holder’s requirements are incompatible with the CC BY 4.0 license, please either i) remove the figure or ii) supply a replacement figure that complies with the CC BY 4.0 license. Please check copyright information on all replacement figures and update the figure caption with source information. If applicable, please specify in the figure caption text when a figure is similar but not identical to the original image and is therefore for illustrative purposes only.

Reviewers' comments:

Reviewer's Responses to Questions

**Comments to the Author**

1. Is the manuscript technically sound, and do the data support the conclusions?

Reviewer #1: Yes

Reviewer #2: Partly

2. Has the statistical analysis been performed appropriately and rigorously? 

Reviewer #1: Yes

Reviewer #2: Yes

3. Have the authors made all data underlying the findings in their manuscript fully available?

Reviewer #1: Yes

Reviewer #2: Yes

4. Is the manuscript presented in an intelligible fashion and written in standard English?

Reviewer #1: Yes

Reviewer #2: Yes

5. Review Comments to the Author

Reviewer #1: The approach and rationale are presented clearly and the mathematics are described in detail. The major concern with this manuscript is that the information presented is already outdated and possibly irrelevant when considering COVID-19. The primary conclusion seems to be that the younger population is not driving the spread of COVID-19 and, thus, the elderly were appropriately targeted first in the vaccine rollouts. I suggest that the authors revise the manuscript to (1) highlight their contributions to the estimation of R0 and considerations of age strata, (2) how their findings perform in the context of COVID-19, and (3) most importantly -- how these techniques/methods are relevant and may be helpful for any future pandemics. Because we are so far into the COVID-19 pandemic and subsequent vaccination strategy, using the methods described here to inform, for instance, vaccine policy recommendations does not seem currently relevant or necessary. Rather, describing how these methods fit the trends seen during the COVID-19 pandemic and translating those findings to applications in future pandemics seems much more relevant.

Reviewer #2: The premise of this article is interesting in that it attempts to marry an analytic method & computing program from computational biology to COVID epidemic reproductive estimation.

The technical outlines and motivations are thorough, and the supplement is much appreciated.

The paper would benefit from a limitations section. First, while the novel application of regime regression to R0 estimation is notable - and the author makes a strong case for it's benefits - there are several confounding variables impacting the results therein. Reporting and testing capacity vary significantly from country to country, and the conclusions should be squared with this reality. In addition, limitations should note that the Prem contact matrices were extrapolated from 8 European countries to 152 nations. The results of R0 estimation from the Prem matrices could be a consequence of this extrapolation.

Lines 407-409: the font appears to be slightly smaller.

The work by the author is impressive, this was thought-provoking to read and review.

6. PLOS authors have the option to publish the peer review history of their article (what does this mean?). If published, this will include your full peer review and any attached files.

Reviewer #1: No

Reviewer #2: No

---

## [Author Response · Author response to Decision Letter 0]

28 May 2021

Please see the uploaded "Response to Reviewers".

---

## [Editor Report · Decision Letter 1]

21 Jun 2021

A comprehensive estimation of country-level basic reproduction numbers R0 for COVID-19: regime regression can automatically estimate the end of the exponential phase in epidemic data

PONE-D-21-06056R1

Dear Dr. Spouge,

We’re pleased to inform you that your manuscript has been judged scientifically suitable for publication and will be formally accepted for publication once it meets all outstanding technical requirements.

Kind regards,

Nicholas S. Duesbery, PhD

Academic Editor

PLOS ONE
---

## [Editor Report · Acceptance letter]

5 Jul 2021

PONE-D-21-06056R1 

A comprehensive estimation of country-level basic reproduction numbers *R*_0_ for COVID-19: regime regression can automatically estimate the end of the exponential phase in epidemic data 

Dear Dr. Spouge:

I'm pleased to inform you that your manuscript has been deemed suitable for publication in PLOS ONE. Congratulations! Your manuscript is now with our production department. 

Kind regards, 

on behalf of

Dr. Nicholas S. Duesbery 

Academic Editor

PLOS ONE